

# Serum proteins differentially expressed in early- and late-onset preeclampsia assessed using iTRAQ proteomics and bioinformatics analyses

Chengcheng Tu[1,*], Feng Tao[1,*], Ying Qin[2], Mingzhu Wu[1], Ji Cheng[1], Min Xie[2], Bing Shen[2], Junjiao Ren[3], Xiaohong Xu[4], Dayan Huang[3] and Hongbo Chen[1]

[1] Department of Obstetrics and Gynecology, Maternal and Child Health Hospital Affiliated to Anhui Medical University, Hefei, Anhui, China

[2] School of Basic Medicine, Anhui Medical University, Hefei, Anhui, China

[3] Department of Science and Education, Maternal and Child Health Hospital Affiliated to Anhui Medical University, Hefei, Anhui, China

[4] Department of Clinical Laboratory, Maternal and Child Health Hospital Affiliated to Anhui Medical University, Hefei, Anhui, China

[*] These authors contributed equally to this work.

Corresponding author
Hongbo Chen,
chenhongbo@ahmu.edu.cn

## ABSTRACT

**Background**. Preeclampsia remains a serious disorder that puts at risk the lives of perinatal mothers and infants worldwide. This study assessed potential pathogenic mechanisms underlying preeclampsia by investigating differentially expressed proteins (DEPs) in the serum of patients with early-onset preeclampsia (EOPE) and late-onset preeclampsia (LOPE) compared with healthy pregnant women.

**Methods**. Blood samples were collected from four women with EOPE, four women with LOPE, and eight women with normal pregnancies, with four women providing control samples for each preeclampsia group. Serum proteins were identified by isobaric tags for relative and absolute quantitation combined with liquid chromatography–tandem mass spectrometry. Serum proteins with differences in their levels compared with control groups of at least 1.2 fold-changes and that were also statistically significantly different between the groups at $P < 0.05$ were further analyzed. Bioinformatics analyses, including gene ontology and Kyoto Encyclopedia of Genes and Genomes signaling pathway analyses, were used to determine the key proteins and signaling pathways associated with the development of PE and to determine those DEPs that differed between women with EOPE and those with LOPE. Key protein identified by mass spectrometry was verified by enzyme linked immunosorbent assay (ELISA).

**Results**. Compared with serum samples from healthy pregnant women, those from women with EOPE displayed 70 proteins that were differentially expressed with significance. Among them, 51 proteins were significantly upregulated and 19 proteins were significantly downregulated. In serum samples from women with LOPE, 24 DEPs were identified , with 10 proteins significantly upregulated and 14 proteins significantly downregulated compared with healthy pregnant women. Bioinformatics analyses indicated that DEPs in both the EOPE and LOPE groups were associated with abnormalities in the activation of the coagulation cascade and complement system as well as with lipid metabolism. In addition, 19 DEPs in the EOPE group were closely related to placental development or invasion of tumor cells. Downregulationof

pregnancy-specific beta-1-glycoprotein 9 (PSG9) in the LOPE group was confirmed by ELISA.

**Conclusion**. The pathogenesis of EOPE and LOPE appeared to be associated with coagulation cascade activation, lipid metabolism, and complement activation. However, the pathogenesis of EOPE also involved processes associated with greater placental injury. This study provided several new proteins in the serum which may be valuable for clinical diagnosis of EOPE and LOPE, and offered potential mechanisms underpinning the development of these disorders.

# INTRODUCTION

Preeclampsia (PE), a hypertensive disorder complicating pregnancy, is the main cause of increased perinatal mortality among mothers and infants worldwide. The rate of PE in the United States is approximately 3.4% (*Ghulmiyyah & Sibai, 2012*), and it is higher in developing countries (*Ananth, Keyes & Wapner, 2013*). PE is marked by new-onset hypertension occurring after 20 weeks of gestation, accompanied by either new-onset proteinuria or systemic multiple organ damage (*Committee on Practice Bulletins—Obstetrics, 2019*). Severe PE may lead to convulsions, coma, cerebral hemorrhage, heart failure, placental abruption, disseminated intravascular coagulation, and even death (*Bibbins-Domingo et al., 2017*). Fetal delivery is the most effective treatment for PE, with other treatments of relieving symptoms used only as an attempt to gain time for enabling further maturity of the fetus. As a result, PE is the leading cause of premature birth and low birth weight, especially severe PE (*Ananth & Vintzileos, 2006*).

The exact pathogenesis of PE remains unclear. However, PE is considered a placenta-derived disease because the syndrome resolves once the placenta is removed. There are two subtypes of PE: early- and late-onset preeclampsia. Their pathogeneses are not identical. Early-onset preeclampsia (EOPE) has a higher degree of placental damage, whereas late-onset preeclampsia (LOPE) may focus on the interaction between the normal senescence of the placenta and a maternal genetic susceptibility to cardiovascular and metabolic disease (*Burton et al., 2019*). However, pathophysiological changes of the placenta further lead to endothelial dysfunction and systemic inflammatory response, which are their common pathogenesis link (*Young, Levine & Karumanchi, 2010*).

The placenta is in direct contact with the maternal circulation. Therefore, any changes in the protein expression of placental tissue are reflected in maternal serum proteins (*Norwitz, 2007*). If the types and levels of differentially expressed proteins (DEPs) in the serum of patients with PE can be researched quantitatively and holistically, specific serum biomarkers may be found for predicting PE and for further study of its pathogenesis. Compared with other proteomics methods, isobaric tags for relative and absolute quantitation (iTRAQ) combined with liquid chromatography–tandem mass spectrometry (LC-MS/MS)

is considered more effective for searching for serum or plasma biomarkers (*Moulder et al., 2018*). Therefore, in the present study, we used iTRAQ and LC-MS/MS methods to identify serum proteins differentially expressed between EOPE and LOPE and between women with PE and healthy pregnant women. We then used bioinformatics analyses to determine the key proteins and related signaling pathways associated with the development of EOPE and LOPE.

## MATERIALS & METHODS

### Participants and clinical samples

The Medical Research Ethics Committee of Anhui Medical University reviewed and approved our research protocol and an informed patient consent form (Anhui Medical Ethics approval No. 20150192). All patients signed the approved written informed consent form prior to being included in the study.

Blood samples were collected from December 2018 to May 2019 from 16 pregnant women, eight of whom had received a diagnosis of PE, and were in the Maternal and Child Health Hospital Affiliated with Anhui Medical University. A PE diagnosis was made consistent with the 2019 American College of Obstetricians and Gynecologists pregnancy hypertension guidance (*Committee on Practice Bulletins—Obstetrics, 2019*). All included pregnant women were primipara and without a disease that may have affected their serum protein levels, such as infection, multiple pregnancy, or gestational diabetes mellitus. Four serum samples were obtained from and allocated to each of the following four groups: EOPE, the EOPE control, LOPE, and the LOPE control groups.

### Protein extraction and quality testing

High-abundance proteins were removed from the serum samples using Pierce Top 12 Abundant Protein Depletion Spin Columns (Thermo Fisher). The protein concentration was determined by the Bradford method. Sodium dodecyl sulfate polyacrylamide gel electrophoresis (SDS-PAGE) was performed to separate the proteins and evaluate the quality of samples.

### Trypsin enzymatic hydrolysis and peptide iTRAQ isobaric labeling

After protein quantification, a centrifuge tube containing 60 μg of protein solution was mixed with 5 μL 1 M dithiothreitol at 37 °C for 1 h. Then 20 μL of 1 M iodoacetamide was added and the solution allowed to react for 1 h at room temperature. All samples were pipetted into ultrafiltration tubes, and the filtrate was discarded after centrifugation. UA buffer (8 M urea, 100 mM Tris-HCl, pH 8.0, 100 μL) was added and the sample centrifuged at 14,000 $g$ for 10 min; this step was repeated twice. Then, 50 mM $NH_4HCO_3$ (50 μL) was added and the filtrate discarded after centrifugation; this step was repeated three times. Trypsin buffer (40 μL) was added and mixed, and the samples were centrifuged at 600 rpm for 1 min. The samples were then subjected to enzymatic hydrolysis at 37 °C for 12–18 h. Transfer the enzymatic hydrolysate to a new centrifuge tube. After labeling with 8plex iTRAQ reagents multiplex kit, the same volume of each sample was mixed together and desalted using a C18 cartridge.

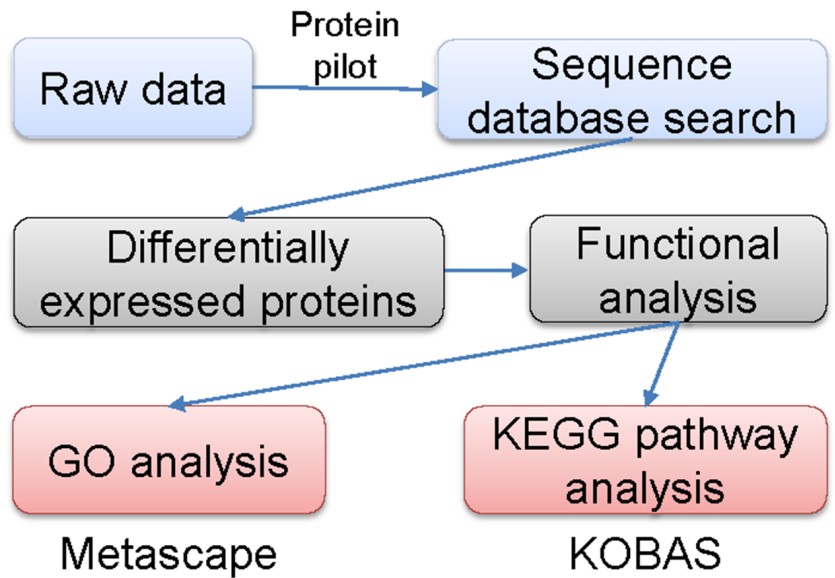

**Figure 1** Flowchart of the data analysis procedure.

## LC-MS/MS analysis

The labeled samples were redissolved in 40 μL of 0.1% formic acid aqueous solution. The peptides were loaded onto a C18-reversed phase column (3 μm C18 resin, 75 μm ×15 cm). The mobile phases consisted of 2% methyl cyanide/0.1% formic acid/98% water and 80% methyl cyanide/0.08% formic acid/20% water. The gradient for the B phase increased linearly at 0–68 min from 7% to 36% and at 68–75 min, from 36% to 100%. Each sample was separated by capillary high-performance liquid chromatography and was analyzed by an Orbitrap Fusion Lumos mass spectrometer (Thermo Science). After the data were collected, they were processed according to the flowchart shown in Fig. 1. Protein identification and quantification were done by using the ProteinPilot Software, version 4.2 (Sciex). Human proteome databases containing UniProt sequences were used to perform peptide identification. Serum proteins with fold changes in their levels compared with control groups of at least 1.2 (in two of two replicates) and that were also statistically significantly different between the groups at $P < 0.05$ were further analyzed.

## Gene ontology (GO) and Kyoto Encyclopedia of Genes and Genomes (KEGG) signaling pathway analyses

Metascape, a web-based resource (http://metascape.org) for gene and protein annotation, visualization, and integration discovery (*Fang et al., 2019*; *Soonthornvacharin et al., 2017*) was used to perform GO analyses. The KOBAS online analysis database (http://kobas.cbi.pku.edu.cn/kobas3) was used to performed KEGG pathway analyses (*Kanehisa & Goto, 2000*). A two-sided $P < 0.05$ was regarded significantly different.
### Enzyme linked immunosorbent assay (ELISA)

PSG9, the downregulated protein in the LOPE group, was verified by ELISA kit (specification and catalogue number: 96T, OM626395). The experiment was performed according to the manufacturer instructions. Blood samples were collected from women with LOPE or normal pregnancies.

### Statistical analysis

Two-tailed Mann–Whitney U test was performed with SigmaPlot software. Values are expressed as means $\pm$ SEM. A value of $P < 0.05$ was considered statistically significant.

## RESULTS

### Participants

The mean and standard deviation (SD) age of the patients in the EOPE group was $31.8 \pm 5.0$ years, and they were at a mean (SD) gestation of $31.9 \pm 2.9$ weeks; the mean (SD) age of the participants in the EOPE control group was $30.0 \pm 1.0$ years, with a mean (SD) gestation of $31.5 \pm 1.4$ weeks. No significant difference was found in age ($P = 0.57$) or gestational weeks ($P = 0.84$) between these two groups. The mean (SD) age of the patients in the LOPE group was $24.8 \pm 1.5$ years, with $37.8 \pm 1.5$ weeks' gestation; the mean (SD) age of the participants in the LOPE control group was $27.0 \pm 1.3$ years, with a mean (SD) of $38.9 \pm 1.3$ weeks' gestation. No significant difference was found in age ($P = 0.15$) or gestation ($P = 0.37$) between these two groups.

### SDS-PAGE

The total proteins in the molecular weight range of 10–220 kDa from 16 samples were effectively separated by SDS-PAGE. The proteins were not degraded, and the high-abundance proteins were obtained (Fig. 2).

### MS/MS spectrum analysis and Identification of DEPs

LC-MS/MS is a powerful tool for identifying proteins in serum samples. We identified 413 serum proteins in the EOPE and EOPE healthy control groups, of which 70 were significantly and differentially expressed between the two groups, with 51 upregulated and 19 downregulated (Table 1, Fig. 3). We also identified 470 proteins in the LOPE and LOPE control groups, of which 24 proteins were significantly and differentially expressed between the two groups, with 10 upregulated and 14 downregulated (Table 2, Fig. 3). Clustergrams generated describing the expression of these DEPs indicated that the expression patterns between the patient EOPE or LOPE groups obviously differed from their control groups, but the two patient groups, EOPE and LOPE, clustered together (Figs. 3B and 3D).

### GO functional annotation and enrichment analysis

GO analysis is an important method and tool in the field of bioinformatics. It includes three categories: cellular component, molecular function, and biological process. GO functional annotation analysis results show the number of DEPs under each item in the three categories. GO functional enrichment analysis provides significant GO functional
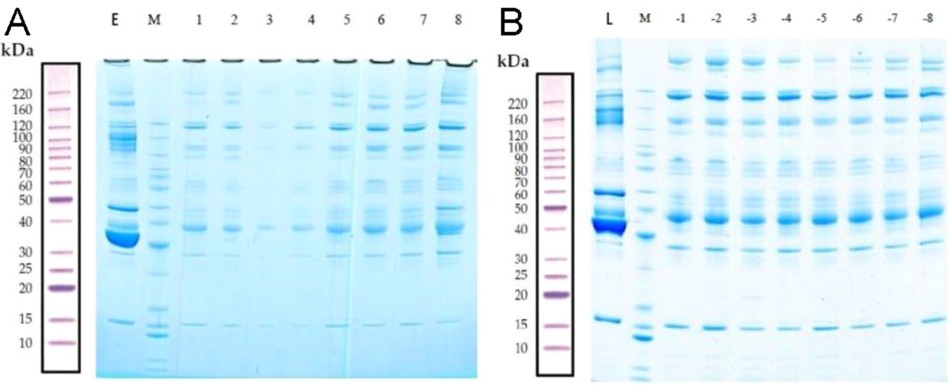

**Figure 2** **Removal of high-abundance proteins.** (A) Lane E is the original sample from the early-onset preeclampsia (EOPE) group. Lane M contains markers. Lanes 1–4 are from serum samples obtained from the EOPE group after removing the highly abundant proteins, whereas lanes 5–8 are from the EOPE control group after removing the highly abundant proteins. (B) Lane L is the original sample from the late-onset preeclampsia (LOPE) group. Lane M contains markers. Lanes −1 to −4 are from the LOPE group after removing the highly abundant proteins, whereas lanes −5 to −8 are from the LOPE control group after removing the high-abundance proteins. The bands are clear and uniform and without protein degradation.

terms associated with the DEPs, that is, those biological functions significantly correlated with the DEP. In vivo, different proteins interact to generate a biological behavior, and a pathway-based analysis helps to further understand those biological functions. A significant pathway enrichment analysis can determine the most important biochemical and metabolic pathways and signal transduction pathways associated with the DEPs.

The GO functional annotation analysis results for the EOPE group are shown in Fig. 4. For the biological process category, the highest percentages of the proteins were associated with the term *biological regulation* ($n = 43$ proteins; with the top three upregulated proteins in this process being CHL1, LRP1, and CNTN1). For the cellular component category, the highest percentages of the proteins were associated with the term *extracellular region* ($n = 49$ proteins; with the top 2 upregulated proteins in this process being MMP2 and CHL1). For the molecular function category, the highest percentages of the proteins were associated with the term *binding* ($n = 45$ proteins; with the top 3 upregulated proteins in this process being MMP2, CHL1, and PEPD). For the LOPE group vs. their controls, the highest percentages of the proteins in the biological process category were associated with the term *biological regulation* ($n = 17$; with the top 3 upregulated proteins in this process being PAPPA2, F7, and vWF). The highest percentages of the proteins in the cellular component category were associated with term *extracellular region* ($n = 18$; with the top three upregulated proteins in this process being PAPPA2, CETP, and F7). The highest percentages of the proteins in the molecular function category were associated with the term *binding* ($n = 13$; with the top 3 upregulated proteins in this process being APPA2, CETP, and F7). The terms in the GO functional enrichment analysis for the EOPE group vs. their controls mainly included response to stress, defense response, and negative regulation of catalytic activity, whereas the terms in the GO functional enrichment analysis
**Table 1** List of differentially expressed proteins in the early-onset preeclampsia group.

| Regulated type | Protein accession | Gene name | Protein description | FC |
|---|---|---|---|---|
| UP | A0A024R6R4 | MMP2 | Matrix metallopeptidase 2 | 21.48 |
| UP | A8K2X4 | - | cDNA FLJ75401 | 12.87 |
| UP | O00533 | CHL1 | Neural cell adhesion molecule L1-like protein | 7.69 |
| UP | J3K000 | PEPD | PEPD protein | 6.6 |
| UP | A8K3I0 | - | cDNA FLJ78437 | 6.31 |
| UP | Q6UXB8 | PI16 | Peptidase inhibitor 16 | 5.76 |
| UP | Q07954 | LRP1 | Prolow-density lipoprotein receptor-related protein 1 | 4.82 |
| UP | Q12860 | CNTN1 | Contactin-1 | 4.64 |
| UP | V9HWB4 | HEL-S-89n | Epididymis secretory sperm binding protein Li 89n | 4.63 |
| UP | P35527 | KRT9 | Keratin, type I cytoskeletal 9 | 4.51 |
| UP | P14543 | NID1 | Nidogen-1 | 3.86 |
| UP | O95236 | APOL3 | Apolipoprotein L3 | 3.4 |
| UP | A6XNE2 | - | Complement factor D preproprotein | 3.34 |
| UP | P41222 | PTGDS | Prostaglandin-H2 | 3.17 |
| UP | V9H1C1 | - | Gelsolin exon 4 (Fragment) | 2.95 |
| UP | Q9Y5Y7 | LYVE1 | Lymphatic vessel endothelial hyaluronic acid receptor 1 | 2.95 |
| UP | Q13201 | MMRN1 | Multimerin-1 | 2.83 |
| UP | B3KQF4 | - | cDNA FLJ90373 | 2.74 |
| UP | Q6MZL2 | DKFZp686M0562 | Uncharacterized protein DKFZp686M0562 (Fragment) | 2.71 |
| UP | J3KPA1 | CRISP3 | Cysteine-rich secretory protein 3 | 2.65 |
| UP | P01034 | CST3 | Cystatin-C | 2.64 |
| UP | A0A0S2Z4F1 | EFEMP1 | EGF Containing Fibulin Extracellular Matrix Protein 1 | 2.62 |
| UP | P23142 | FBLN1 | Fibulin-1 | 2.57 |
| UP | A0A384N669 | - | Epididymis secretory sperm binding protein | 2.48 |
| UP | J3KNB4 | CAMP | Cathelicidin antimicrobial peptide | 2.47 |
| UP | B7Z544 | - | cDNA FLJ51742 | 2.43 |
| UP | A8K061 | - | cDNA FLJ77880 | 2.42 |
| UP | A0A087WV75 | NCAM1 | Neural cell adhesion molecule 1 | 2.37 |
| UP | Q9NQ79 | CRTAC1 | Cartilage acidic protein 1 | 2.24 |
| UP | B2RBW9 | - | cDNA, FLJ95746 | 2.23 |
| UP | A0A024RAA7 | C1QC | Adiponectin B | 2.15 |
| UP | P16070 | CD44 | CD44 antigen | 2.06 2.01 |
| UP | Q76LX8 | ADAMTS13 | Von Willebrand Factor-Cleaving Protease | 2.01 |
| UP | Q6EMK4 | VASN | Vasorin | 2 |
| UP | D6RE86 | CP | Ceruloplasmin (Fragment) | 1.88 |
| UP | H6VRG1 | KRT1 | Keratin 1 | 1.87 |
| UP | Q86U17 | SERPINA11 | Serpin A11 | 1.81 |
| UP | Q9UKE5 | TNIK | TRAF2 and NCK-interacting protein kinase | 1.75 |
| UP | P61626 | LYZ | Lysozyme C | 1.74 |
| UP | P02743 | APCS | Serum amyloid P-component | 1.74 |

**Table 1** (*continued*)

| Regulated type | Protein accession | Gene name | Protein description | FC |
|---|---|---|---|---|
| UP | Q16853 | AOC3 | Membrane primary amine oxidase | 1.73 |
| UP | Q9NZP8 | C1RL | Complement C1r subcomponent-like protein | 1.73 |
| UP | A0A024R035 | C9 | Complement C9 | 1.71 |
| UP | A0A0A0MRJ7 | F5 | Coagulation factor V | 1.66 |
| UP | P09871 | C1S | Complement C1s subcomponent | 1.65 |
| UP | Q6PIL8 | IGK@ | IGK@ protein | 1.63 |
| UP | P49908 | SELENOP | Selenoprotein P | 1.55 |
| UP | P02654 | APOC1 | Apolipoprotein C-I | 1.55 |
| UP | D3DRR6 | ITIH2 | Inter-alpha (Globulin) inhibitor H2 | 1.54 |
| UP | B4DPQ0 | C1R | Complement C1r subcomponent | 1.52 |
| UP | J3KNP4 | SEMA4B | Semaphorin-4B | 1.51 |
| DOWN | Q6GMX6 | IGH@ | IGH@ protein | 0.56 |
| DOWN | A0A0X9TD47 | - | MS-D1 light chain variable region (Fragment) | 0.43 |
| DOWN | Q9UNU2 | C4B | Complement protein C4B frameshift mutant (Fragment) | 0.43 |
| DOWN | P01591 | JCHAIN | Immunoglobulin J chain | 0.34 |
| DOWN | Q14213 | EBI3 | Interleukin-27 subunit beta | 0.34 |
| DOWN | P28799 | GRN | Granulins | 0.33 |
| DOWN | B2R7N9 | - | cDNA, FLJ93532 | 0.32 |
| DOWN | P00709 | LALBA | Alpha-lactalbumin | 0.29 |
| DOWN | G5E9F7 | PSG1 | Pregnancy specific beta-1-glycoprotein 1 | 0.25 |
| DOWN | A0A075B6A0 | IGHM | Immunoglobulin heavy constant mu (Fragment) | 0.23 |
| DOWN | A0A075B6R9 | IGKV2D-24 | Immunoglobulin kappa variable 2D-24 | 0.23 |
| DOWN | O75636 | FCN3 | Ficolin-3 | 0.22 |
| DOWN | O43866 | CD5L | CD5 antigen-like | 0.2 |
| DOWN | P06733 | ENO1 | Alpha-enolase | 0.16 |
| DOWN | P09172 | DBH | Dopamine beta-hydroxylase | 0.16 |
| DOWN | Q15485 | FCN2 | Ficolin-2 | 0.13 |
| DOWN | Q86TT1 | | Full-length cDNA clone CS0DD006YL02 of Neuroblastoma | 0.08 |
| DOWN | P00739 | HPR | Haptoglobin-related protein | 0.06 |
| DOWN | P00738 | HP | Haptoglobin | 0.01 |

for the LOPE group vs. their controls included enzyme inhibitor activity and serine-type endopeptidase inhibitor activity (Fig. 5).

## KEGG signaling pathway analysis

The KOBAS online analysis tool was used to identify the functions associated with the DEPs and the KEGG signaling pathways. Results of the KEGG pathway enrichment analysis showed that complement and coagulation cascades, proteoglycans in cancer, and metabolic pathways were the main signaling pathways associated with the EOPE group, whereas complement and coagulation cascades were the main signaling pathways associated with the LOPE group (Fig. 6). Thus, the results indicated that a functional change in coagulation was the main finding for both EOPE and LOPE.

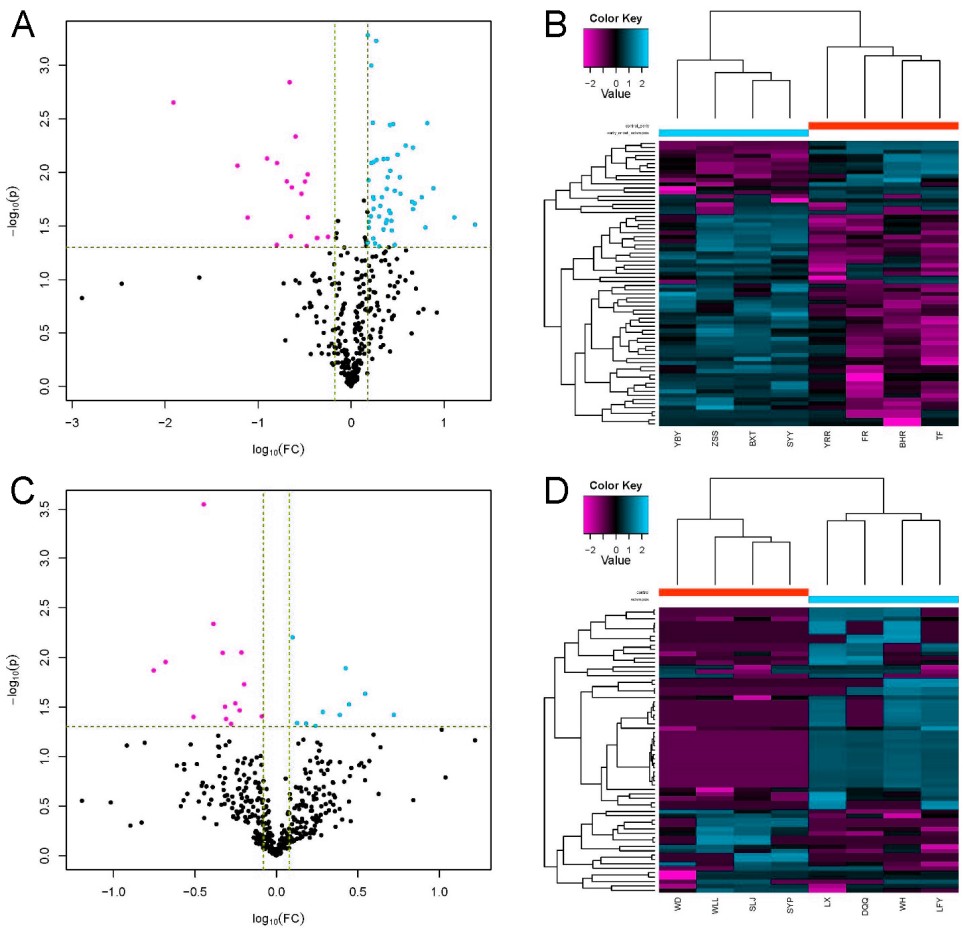

**Figure 3  Differential protein expression.** (A and C) Volcano plots with red dots on the right-hand side indicating upregulation, green dots on the left-hand side indicating downregulation, and black dots indicating no significant change in protein expression levels based on the criteria of an absolute log10 fold change and $P < 0.05$ between early-onset preeclampsia and its control group (A), and between late-onset preeclampsia and its control group (C). (B and D) Clustergram for the expression of the DEPs between early-onset preeclampsia and control (B) and between late-onset preeclampsia and control (D).

### PSG9 protein level in serum

Serum level of PSG9 was measured by ELISA. The data showed a significant difference between LOPE group and control group ($4.54 \pm 1.22$ vs. $6.32 \pm 1.73$, Fig. 7). The result is consistent with the study of mass spectrometry.

## DISCUSSION

Proteomics methods have provided some important information regarding PE. For example, *Blumenstein et al. (2009)* used a differential in gel electrophoresis–based approach to identify changes in the plasma proteome of pregnant women who subsequently developed PE. They found that those DEPs are mainly involved in lipoprotein metabolism, the blood coagulation system, and the complement system. The results of the present study were

**Table 2 List of differentially expressed proteins in the late-onset preeclampsia group.**

| Regulated type | Protein accession | GENE NAME | PROTEIN DESCRIPTION | FC |
|---|---|---|---|---|
| UP | Q9BXP8 | PAPPA2 | Pappalysin-2 | 5.27 |
| UP | A0A0S2Z3F6 | CETP | Cholesteryl ester transfer protein plasma isoform 1 | 3.51 |
| UP | P08709 | F7 | Coagulation factor VII | 2.8 |
| UP | Q6NS95 | IGL@ | IGL@ protein | 2.67 |
| UP | P04275 | VWF | von Willebrand factor | 2.46 |
| UP | C0JYY2 | APOB | Apolipoprotein B | 1.93 |
| UP | A0A140VK24 | - | Testicular secretory protein Li 24 protein 1 | 1.74 |
| UP | P05546 | SERPIND1 | Heparin cofactor 2 | 1.52 |
| UP | O95445 | APOM | Apolipoprotein M | 1.34 |
| UP | P80108 | GPLD1 | Phosphatidylinositol-glycan-specific phospholipase D | 1.26 |
| DOWN | P05543 | SERPINA7 | Thyroxine-binding globulin | 0.82 |
| DOWN | Q9NZP8 | C1RL | Complement C1r subcomponent-like protein | 0.63 |
| DOWN | A8K2T7 | - | Receptor protein-tyrosine kinase | 0.61 |
| DOWN | Q00887 | PSG9 | Pregnancy-specific beta-1-glycoprotein 9 | 0.6 |
| DOWN | P07333 | CSF1R | Macrophage colony-stimulating factor 1 receptor | 0.56 |
| DOWN | Q2L9S7 | AAT | Alpha-1-antitrypsin MBrescia variant receptor 1 | 0.53 |
| DOWN | A0A2S0BDD1 | ATIII-R2 | Antithrombin III isoform | 0.49 |
| DOWN | Q8WW79 | SELL | L-selectin | 0.48 |
| DOWN | P40189 | IL6ST | Interleukin-6 receptor subunit beta (Fragment) | 0.47 |
| DOWN | B2R7Y0 | - | cDNA, FLJ93654 | 0.41 |
| DOWN | Q12860 | CNTN1 | member 2 (SERPINB2) | 0.36 |
| DOWN | Q8WWZ8 | OIT3 | Oncoprotein-induced transcript 3 protein | 0.31 |
| DOWN | A0A140TA29 | C4B | Complement C4-B | 0.21 |
| DOWN | B2R950 | - | cDNA, FLJ94213 | 0.18 |

consistent with those of *Blumenstein et al. (2009)*, but we also made some new discoveries (discussed below). These results suggest that the pathogenesis of PE is complex and is associated with multiple proteins and signaling pathways.

To maintain pregnancy and fetal growth progress within normal reference ranges, healthy pregnant women develop physiological hyperlipidemia. However, they do not experience angiopathy because high-density lipoprotein (a vascular protection factor) and low-density lipoprotein (LDL, an atherogenic factor) also increase to protect the vascular endothelium from injury (*Jin et al., 2016*; *Wang et al., 2017*). Compared with healthy pregnant women, pregnant women with PE display significantly increased serum triglycerides and LDLs, which may enhance oxidative stress and ultimately lead to vascular endothelial cell injury (*Huda, Sattar & Freeman, 2009*; *Pohanka, 2013*). Our study found that there were six DEPs associated with lipid regulation increased in the PE groups, including ApoB, LRP1, ApoL3, ApoC-I, CETP, and ApoM. ApoB, the main carrier protein of LDL, is an atherogenic risk factor. Previous evidence supports that ApoB is significantly upregulated in the plasma of patients with PE (*Lin et al., 2019*). CETP plays an important role in high-density lipoprotein metabolism and reverse cholesterol transport and is upregulated in the third trimester of pregnancy. Another study has reported that the TaqIB

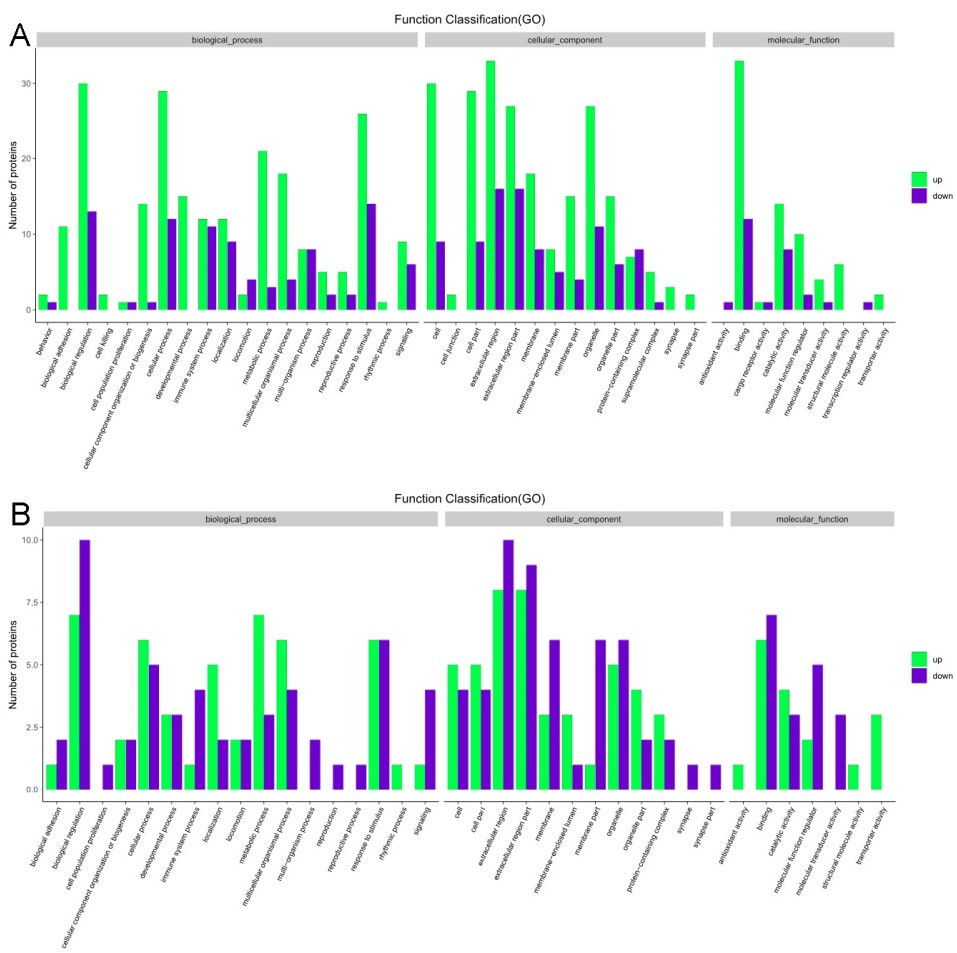

**Figure 4** **Gene ontology functional annotation.** The upregulated and downregulated proteins in the three categories of biological process, cellular component, and molecular function in (A) the early-onset preeclampsia group vs. its control group and (B) the late-onset preeclampsia group vs. its control group.

polymorphism of the *CETP* gene is significantly correlated with the triglyceride and total cholesterol levels in patients with severe PE (*Belo et al., 2004*). Here, we reported for the first time, to our knowledge, that LRP1, ApoC1, and ApoL3 are increased in the EOPE group. LRP1 and ApoC1 accelerate the development of atherosclerosis through different pathways (*Mueller et al., 2018*; *Westerterp et al., 2007*). Endothelial function decreases during PE, and endothelial dysfunction is a characteristic of atherosclerosis. ApoL3, as a regulator of MAPK and FAK signaling in endothelial cells, has been shown to be involved in angiogenesis in vitro (*Khalil et al., 2018*). Increased ApoL3 in PE may be a compensatory response to endothelial dysfunction. Therefore, the current evidence suggests that dyslipidemia may be related to the development of PE, but the specific pathogenesis remains to be explored in future studies.

Physiological hypercoagulability in healthy pregnant women can prevent intrapartum and postpartum hemorrhage. However, abnormalities in coagulation, the anti-coagulation

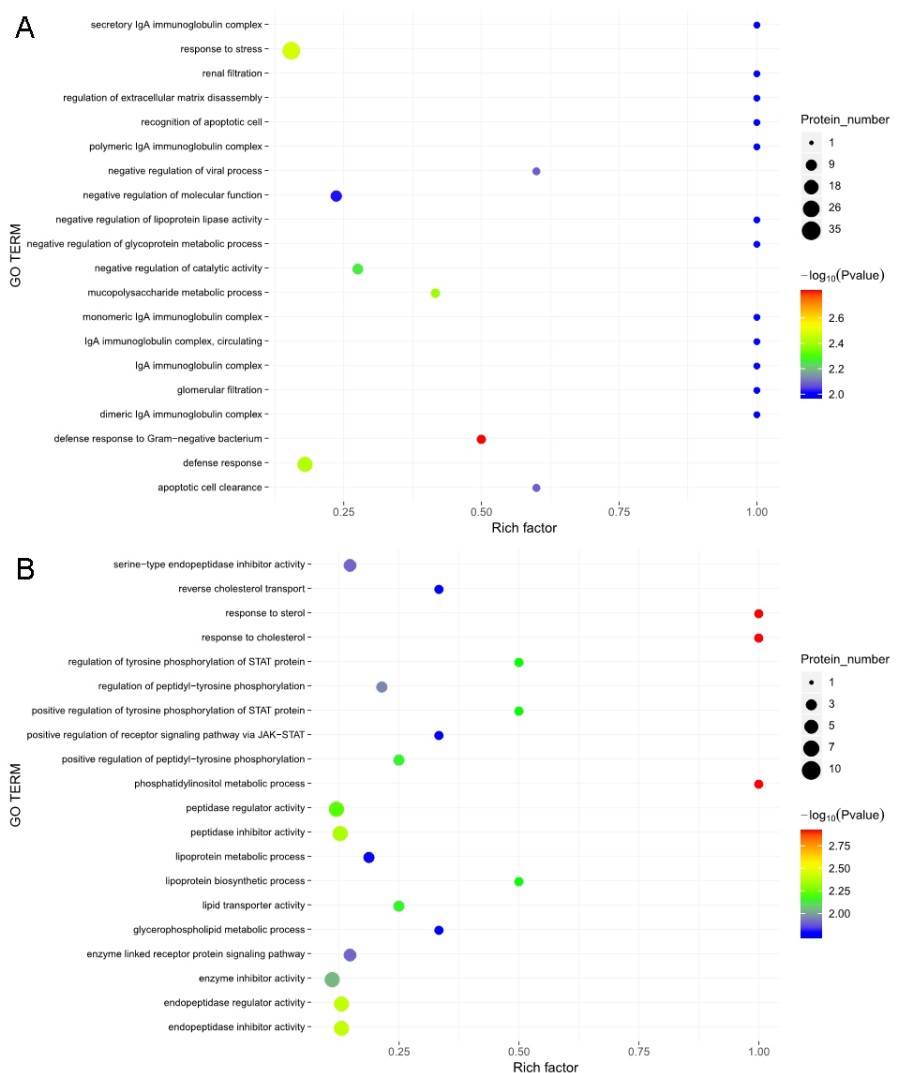

**Figure 5  Gene ontology (GO) functional enrichment.** Enriched GO terms for the upregulated and downregulated proteins and the associated protein numbers in (A) early-onset preeclampsia vs. its control and (B) late-onset preeclampsia vs. its control.

system, and the fibrinolytic system lead to pathological hypercoagulability in PE (*Dusse et al., 2011*). In the LOPE group of the present study, coagulation factors VII, vWF, and SRPIND1 were upregulated group, whereas AAT and ATIII were downregulated compared with healthy controls. In the EOPE group, coagulation factor V, plasminogen, and ADAMTS13 were upregulated. These results are consistent with those of many previous trials (discussed below). The level of plasma FVII in patients with severe PE is significantly higher than that in healthy pregnant women; therefore plasma FVII levels may show high sensitivity and specificity in differentiating between PE and normal pregnancy (*Dusse et al., 2016*). The coagulation factor vWF is a specific marker that reflects damage of endothelial cells; thus, damaged microvascular endothelial cells in PE promote the expression of

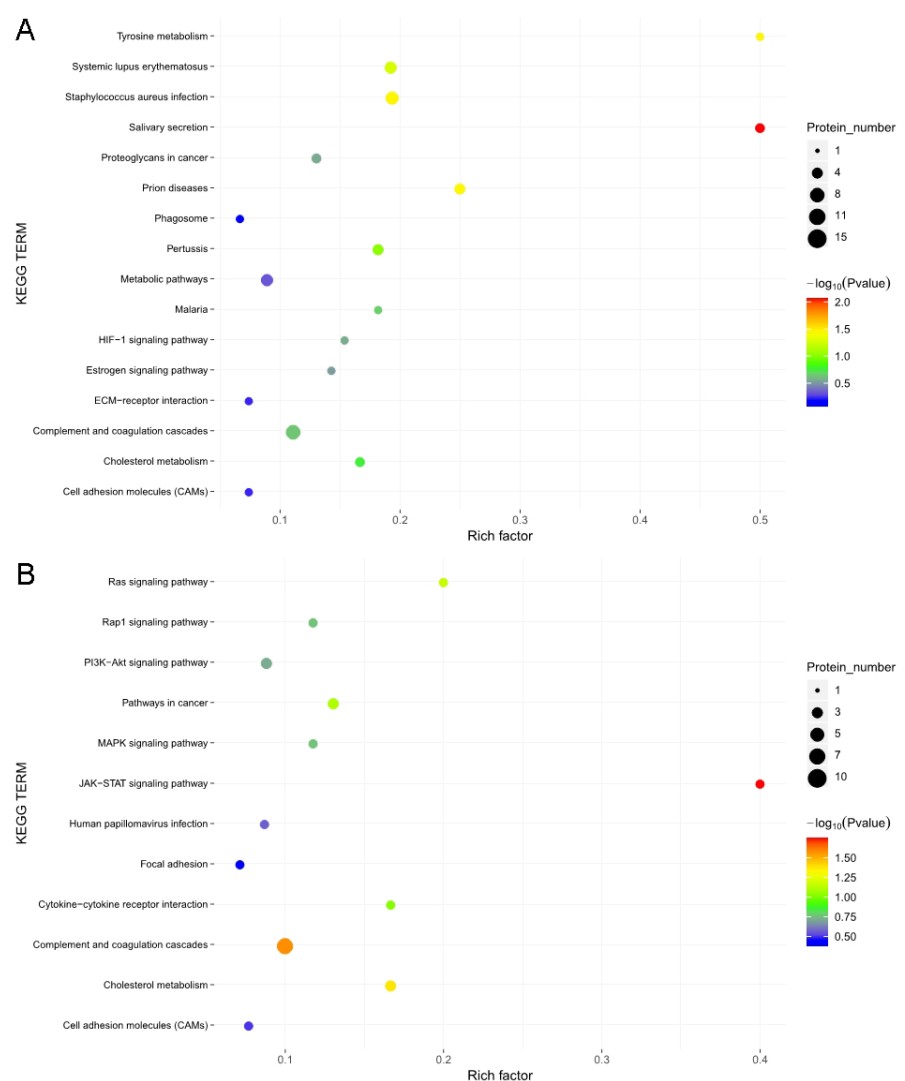

**Figure 6  Kyoto Encyclopedia of Genes and Genomes (KEGG) pathway enrichment.** KEGG pathways enriched for the upregulated and downregulated proteins and the associated protein numbers in (A) early-onset preeclampsia vs. its control and (B) late-onset preeclampsia vs. its control.

vWF. Owing to the activation of the intrinsic and exogenous coagulation pathways by damaged endothelial cells, a large number of coagulation factors are activated. This leads to a massive generation of thrombin, which antagonizes a large amount of anti-thrombin III, resulting in a significant decrease in its level (*Demir & Dilek, 2010*; *Pottecher et al., 2009*). In response to abnormalities in the coagulation mechanism of PE, evidence-based medical study has also shown that oral administration of low-dose aspirin during early pregnancy can significantly reduce the incidence rate of PE (*Rolnik et al., 2017*).

A growing number of studies have shown that abnormal expression with the complement system is associated with PE (*Agostinis et al., 2010*; *Derzsy et al., 2010*). Our results support this evidence, showing that expression levels of CFD, CIQC, CIRL, C9, C1S, and C1R were

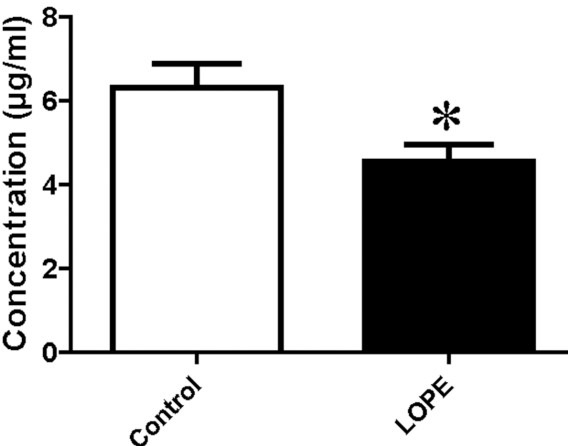

**Figure 7 Protein level of pregnancy-specific beta-1-glycoprotein 9 (PSG9) in serum.** Concentration of PSG9 in the patient serum in control and LOPE groups. Values are shown as the mean ±SEM (n = 10); *P < 0.05 for Control vs. LOPE.

significantly increased, while expression levels of C4B, FCN2, and FCN3 were significantly decreased in the EOPE group compared with their control group. Moreover, the expression levels of complement C1RL and C4B were decreased in the LOPE group compared with their controls.

Downregulation of PSG1 and PSG9 expressed in the PE groups is an interesting finding in our study. PSG is a pregnancy-specific glycoprotein that is synthesized and secreted into the blood by placental syncytiotrophoblast cells. *Rong et al. (2017)* found that PSG9 significantly promotes the angiogenesis of human umbilical vein endothelial cells. *Chang et al. (2016)* found a significant increase in the number of cases with deletions in the PSG gene locus among patients with PE. We hypothesize that downregulation of the PSG protein family may be involved in the pathogenesis of PE by affecting the proliferation of endothelial cells although no specific experimental evidence currently supports or disproves this hypothesis.

PE is considered a placenta-derived disease. In the classic two-stage model, placental stress leads to dysfunction of maternal peripheral endothelial cells, systemic inflammatory response, and the clinical syndrome of PE (*Staff, 2019*). This model is also reflected in our mass spectrometry results. In both the EOPE and LOPE groups, abnormalities of coagulation cascade activation, lipid metabolism, and complement activation were found. However, the main stressor associated with EOPE is placental hypoperfusion secondary to impaired spiral artery remodeling; by contrast, in LOPE, the cause is more likely attributable to a mismatch between normal maternal perfusion and the metabolic demands of the placenta and fetus (*Burton et al., 2019*). Spiral artery remodeling is completed by the placental trophoblast cells continuously invading and destroying the uterine spiral arterial wall and gradually replacing the endothelial cells. Many studies have focused on the control of this invasion (*Pollheimer et al., 2018*; *Wagner, Otomo & Christians, 2011*; *Wu et al., 2016*).

Unexpectedly, we also found 19 DEPs in the EOPE group that were closely related to placental development or invasion of tumor cells. This result was not detected in the LOPE group, which showed an increase only in PAPPA2, which affects the invasion and metastasis of placental trophoblasts and leads to a gradual decline in placental function (*Wagner, Otomo & Christians, 2011*). LYVE1, CST3, and NCAM1 are also reportedly involved in placental vascular remodeling (*Pawlak et al., 2019*; *Song et al., 2010*; *Zhang, Xu & Han, 2019*). GRN, CD5L, ENO1, CHL1, CRISP3, VASN, and TNIK promote the invasive ability of tumor cells (*Aran et al., 2018*; *Bhandari et al., 2019*; *Buhusi et al., 2003*; *Chon et al., 2016*; *Song et al., 2014*; *Voshtani et al., 2019*; *Wang et al., 2019*). By contrast, SEMA4B, KRT1, KRT9, FBLN1, and PTGDS are involved in the anti-invasive activity of tumor cells (*Blanckaert et al., 2015*; *Jian et al., 2015*; *Marano et al., 2018*; *Zhang et al., 2018*). MMP2 plays an important role in trophoblast invasion and is generally thought to be downregulated in PE (*Wu et al., 2016*). We found the opposite result, that is, the expression of MMP2 was upregulated in EOPE; however, expression of PI16, which inhibits MMP2 activity (*Hazell et al., 2016*), was also upregulated. PEPD is a hydrolase that affects collagen biosynthesis, cell proliferation, and matrix remodeling. *Pehlivan et al. (2017)* have shown that PEPD activity in the plasma, umbilical cord, and placental tissue of women with pregnancy-induced hypertension is higher than that of healthy pregnant women. Our results also indicated an upregulation of PEPD in the EOPE group. The dysregulation of the expression of these proteins may be an important cause of placental dysfunction in EOPE.

Furthermore, previous studies have shown that soluble form of placenta-derived endoglin (sENG) was upregulated in the serum of EOPE, which is involved in endothelial dysfunction in coordination with soluble fms-like tyrosine kinase (*Venkatesha et al., 2006*). Our mass spectrometry results were similar to that previous finding showing that endoglin was upregulated in EOPE but not in LOPE group. This finding may also indicate more severe endothelial dysfunction in EOPE.

# CONCLUSIONS

In summary, the use of iTRAQ combined with LC-MS/MS was effective for screening serum for DEPs in PE. We used bioinformatics to analyze the DEPs that showed significant changes in their expression levels to provide potential indicators for detecting PE and ideas for the study of pathogenesis in PE. In conclusion, the pathogenesis of EOPE and LOPE appeared to be related to dysfunctions in coagulation cascade activation, lipid metabolism, and complement activation. However, compared with that in LOPE, the pathogenesis of EOPE was associated with greater placental injury. This study provides several new proteins in the serum which may potentially have value in clinical diagnosis of EOPE and LOPE and offers potential mechanisms for the development of EOPE and LOPE, contributing to the future research on prediction, prevention, and pathogenesis of PE.

### Funding

This work was supported by the Anhui Province Science and Technology Innovation Project Demonstration Project (No. 201707d08050003), and the Anhui Province Key Research and Development Project (No. 201904a07020032). The funders had no role in study design, data collection and analysis, decision to publish, or preparation of the manuscript.

### Grant Disclosures

The following grant information was disclosed by the authors:
Anhui Province Science and Technology Innovation Project Demonstration Project: 201707d08050003.
The Anhui Province Key Research and Development Project: 201904a07020032.

### Competing Interests

The authors declare there are no competing interests.

### Author Contributions

- Chengcheng Tu, Feng Tao and Ying Qin performed the experiments, prepared figures and/or tables, and approved the final draft.
- Mingzhu Wu and Dayan Huang analyzed the data, prepared figures and/or tables, authored or reviewed drafts of the paper, and approved the final draft.
- Ji Cheng analyzed the data, authored or reviewed drafts of the paper, and approved the final draft.
- Min Xie performed the experiments, authored or reviewed drafts of the paper, and approved the final draft.
- Bing Shen conceived and designed the experiments, prepared figures and/or tables, authored or reviewed drafts of the paper, and approved the final draft.
- Junjiao Ren analyzed the data, prepared figures and/or tables, and approved the final draft.
- Xiaohong Xu performed the experiments, analyzed the data, prepared figures and/or tables, and approved the final draft.
- Hongbo Chen conceived and designed the experiments, authored or reviewed drafts of the paper, and approved the final draft.

### Human Ethics

The following information was supplied relating to ethical approvals (i.e., approving body and any reference numbers):

The Medical Research Ethics Committee of Anhui Medical University reviewed and approved our research protocol and an informed patient consent form (Anhui Medical Ethics approval No. 20150192).

## Ethics

The following information was supplied relating to ethical approvals (i.e., approving body and any reference numbers):

The Medical Research Ethics Committee of Anhui Medical University reviewed and approved our research protocol and an informed patient consent form (Anhui Medical Ethics approval No. 20150192).

## Data Availability

The raw measurements are available as Supplemental Files.

## Supplemental Information

Supplemental information for this article can be found online at http://dx.doi.org/10.7717/peerj.9753#supplemental-information.

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
