# Peer review of "Serum proteins differentially expressed in early- and late-onset preeclampsia assessed using iTRAQ proteomics and bioinformatics analyses"

_PeerJ, doi:10.7717/peerj.9753_

## Round 0.1 · original submission · Major Revisions

Three specialists in the field evaluated your manuscripts. Two of them recommend major revisions. I agree with them - please respond to all the comments.

Reviewer 1 ·

Basic reporting

No comment.

Experimental design

The use of small sample (4 per sub-group) size should be justified.

Validity of the findings

1. Were the top three differentially expressed proteins equal in all the samples of the sub-groups?

2. LRP1, APOC1, and APOL3 were linked to which gestational sub-group?

3. Whether the differentially expressed proteins are placenta-specific needs be clarified. If not, how do you justify them as a biomarker?

4. Power analysis should be carried out to determine the acceptability of the findings.

Reviewer 2 ·

Basic reporting

In this manuscript, the authors analyze a set of serum samples obtained either from control individuals or patients diagnosed with early and late preeclampsia. Differential quantitative proteomics experiments have been carried out using iTRAQ 8-plex isobaric labeling as an experimental approach, followed by LC-MS analysis of the samples. The main objective of this work is to describe proteins differentially expressed between the control and problem (preeclampsia) samples in order to find new clues related to this pathology. This is an objective of great biomedical interest, since preeclampsia has a great impact on the human population, causing high economic and human costs, and for which clear and unambiguous biomarkers useful for diagnosis and prognosis have not yet been found. Both the introduction and, above all, the discussion of the results adequately pose the problem and the experimental context. In addition, the manuscript is well written and easy to understand. For everything mentioned above, it is a manuscript of potential interest for publication. However, significant improvements must be done, particularly in the data presentation format, as described below.

Experimental design

The experimental design proposed by the authors is not novel, but it is adequate to try to obtain the answers sought, although it could be objected that the number of samples used and representative of each condition is somewhat low. Likewise, it would be advisable to include some experiments to validate at least some of the so-called differentially expressed proteins identified here, either using MRM directed proteomic techniques or other approaches based on the use of antibodies, such as Western Blot analysis. The validation should ideally be performed on a cohort of samples not related to that used in the first phase.

Validity of the findings

MAJOR POINTS:
The presentation of the experimental data requires further improvement. For example, in the results file corresponding to LOPE, 8 data columns (supposedly corresponding to 4 control samples + 4 LOPE samples) are displayed. However, in the EOPE file, 16 data columns are shown, and apparently it includes all (8) the control samples and both 4 LOPE samples + 4 EOPE samples. Please modify this as it only serves to create confusion. In addition, the authors should indicate in the table which samples correspond either to the control or preeclampsia groups.

Lines 135-140: Data processing and protein identification and quantification analysis methodology is confusing and contains some spelling mistakes, for example, writing Unipro instead of UniProt. It should be explained why using ProteinPilot to analyze the results instead of using other alternatives recommended by Thermo (please note that data were obtained in a Thermo Orbitrap Fusion Lumos mass spectrometer) such as ProteomeDiscoverer. Other common alternatives, and by far more used by the proteomics community, include freely available software like MaxQuant.

In addition, there is no coincidence when the proteins found in the result files and the protein numbers mentioned in the text are compared. Thus, 65 and 132 proteins, respectively, are shown in LOPE and EOPE files, values that do not match the total number of proteins identified in both data sets (413 for EOPE, 470 for LOPE) or the proteins described as differentially regulated (70 and 24 for EOPE and LOPE, respectively). I do not understand why quantification values corresponding to all the proteins, either differentially expressed or not, identified in both data groups (EOPE and LOPE) are not shown. I suggest that the authors present the data so that we can:
1) Have the quantification values of all the proteins identified in each data set.
2) highlight the values corresponding to differentially regulated proteins in each data set independently.

Regarding the quantification values presented in the table, it is well known that the number of decimal numbers should reflect the accuracy of the technique used. It is not a proper practice to present the data with such a high number of decimal numbers, since differential quantitative proteomic analysis using iTRAQ 8plex is far from reaching such high accuracy. I suggest the authors to use 2 or 3 decimal numbers in the tables, because it will help to better understand the data.

MINOR POINTS:

Line 118: dithiothreitol final concentration
Line 118: iodoacetamide final concentration
Line 120: UA? Please detail
Iine 125: 75 micrograms of sample are obtained but the protocol starts with 60 micrograms (line 117): please clarify this point

Discussion:
I was surprised that the authors did not include in the discussion any mention to the protein “cDNA FLJ75401, highly similar to Homo sapiens endoglin”, described as differentially expressed protein in the EOPE dataset, since this protein has been previously described as a putative preeclampsia marker (Venkatesha S, Toporsian M, Lam C, Hanai J, Mammoto T, Kim YM, Bdolah Y, Lim KH, Yuan HT, Libermann TA, Stillman IE, Roberts D, D'Amore PA, Epstein FH, Sellke FW, Romero R, Sukhatme VP, Letarte M, Karumanchi SA. Soluble endoglin contributes to the pathogenesis of preeclampsia Nat. Med. 2006 Jun; 12 (6): 642-9.). Particularly interesting would be to discuss the presence of this protein in the EOPE but not in the LOPE dataset.

Additional comments

no comment

·

Basic reporting

The manuscript entitled “Serum proteins differentially expressed in early- and late-onset preeclampsia assessed using iTRAQ proteomics and bioinformatics analyses” by Tu and his co-authors reported the differently expressed serum proteins in early- and late-onset preeclampsia. The manuscript reported some new findings that are associated with preeclampsia.

The English language used is a professional standard.
Literature and references are sufficient.
Article structure, figures, and tables are sufficient and standard.

Experimental design

The only limitation is the small number of samples.

Validity of the findings

The data provided are robust and statistically sound. The authors explained their findings in a proper way.

Additional comments

The manuscript entitled “Serum proteins differentially expressed in early- and late-onset preeclampsia assessed using iTRAQ proteomics and bioinformatics analyses” by Tu and his co-authors reported the differently expressed serum proteins in early- and late-onset preeclampsia. The manuscript reported some new findings that are associated with preeclampsia.

---

## Round 0.2 · Minor Revisions

Two reviewers evaluated the revised version of this manuscript. Both reviewers asked for additional corrections. In my view, this manuscript needs a minor revision.

Reviewer 1 ·

Basic reporting

See attachment

Experimental design

See attachment

Validity of the findings

See attachment

Additional comments

See attachment

Annotated reviews are not available for download in order to protect the identity of reviewers who chose to remain anonymous.

Reviewer 2 ·

Basic reporting

Please see my first report

Experimental design

Please see my first report

Validity of the findings

Please see my first report

Additional comments

The authors have made an effort to introduce the changes and modifications suggested in the previous version. As a consequence, I understand that the overall quality of the manuscript has improved substantially and that now it can be considered acceptable for publication. I would just like to suggest a few minor changes to the text that are detailed below.
Line 33: What is exactly the meaning of the expression “an absolute log10 fold change (i.e., >1.2 difference)”? It seems confusing to me. Could it be expressed more clearly?
Line 48: Use Downregulation instead of “The downregulated”
Line 83: What does it mean R 2007?
Line 127: The expression “All the remaining samples were used” is unclear
Line 128: Use "After labeling with iTRAQ" instead of "After labeling". In addition it should be specified the iTRAQ version (4plex? 8plex?) across the manuscript.
Line 134-135: methyl cyanide is correct but very few people use it. It is much popular to use acetonitrile.
Line 155: please specify the reference of the kit used
Line 156: use "manufacturer instructions" instead of "kit instructions"
Line 197: It is more appropriate to use "interact" instead of writing "coordinate with one another"
Line 231: the expression "we used ELISA to identify again" is grammatically wrong, but it is also meaningless. Please correct.
Line 255: Use "previous evidence" instead of "much evidence"
Line 278: Factor VII instead of factor VI?
Line 284:" consumes" is inappropriate in this context, please use a more adequate expression.
Line 291: Use "support" instead of "add to"
Line 339: please use "previous studies have shown that" instead of "a study shows that"
Line 340: please use "which is involved" instead of "which involved"
line 342: please use "were similar" instead of "was similar"

---

## Round 0.3 · accepted · Accept

I would like to start this letter with an apology for the delay in my decision. I'm sure you understand that everybody's lives have been disrupted by the pandemic, which has both taken us away from our offices and changed the timing and length of our (remote) meetings. I really wanted to read the manuscript very carefully and assess whether you had revised the manuscript in line with the reviewer's comments. After a careful reading I have reached the conclusion that you have done so; therefore, I feel this version is now acceptable.

Please accept my apologies. I hope that things are going well for you during these trying times.